# Pathway towards a High Recycling Content in Traditional Ceramics

Elisa Rambaldi 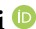

Centro Ceramico, Via Martelli 26, 40138 Bologna, Italy; rambaldi@centroceramico.it; Tel.: +39-051534015

**Abstract:** The present work shows the path towards the industrial production of ceramic tiles containing a high amount of recycling materials in the substitution of natural raw materials. Starting from the applied research at laboratory scale, which is able to demonstrate the work feasibility, other important milestones consist of pilot scale production until the proper industrial production. Finally, when all these steps are positively achieved, the practice is consolidated and it is possible to reach the concrete sustainability benefits (social, environmental and economic). The results of an industry driven project that aimed to produce porcelain stoneware tiles containing 85% of recycled materials were selected to show this path. This innovative ceramic product—containing soda-lime scrap glass from urban-separated collection (post-consumer waste) and unfired scrap tiles from industrial ceramic process (pre-consumer waste)—was sintered about 200 °C lower than a traditional porcelain stoneware tile. It maintains high technical performances belonging to class BIa of the International Standard of ceramic tile classification (EN ISO 14411). Moreover, this product fulfils the standard requirements for dry-pressed ceramic tiles with low water absorption ($\leq$0.5%), and it obtained the certification UNI Keymark. The LCA study was also performed and the results showed a significantly lower environmental impact of this innovative product compared to a traditional porcelain stoneware tile.

**Keywords:** porcelain stoneware tiles; sustainability; recycling; technical performances; certifications

## 1. Introduction

Today, traditional ceramic tiles are already marketed with characteristics and performances that go far beyond traditional uses. The driver of these innovations (photocatalysis, antibacterial properties, high reflectance index, etc.) is sustainability [1–4]. These are not just future possibilities because their industrial and commercial reality already make them serviceable in multiple environments.

Sustainability is also the driver for technological innovation (shaping machines, kilns, digital printing, etc.) able to produce high quality ceramic tiles in several sizes—from the traditional 30 cm × 30 cm to the last large format of 3 m × 1.5 m and with thicknesses ranging from 3 mm to 3 cm, and high level aesthetic appearance able to simulate natural materials such as wood, marble, textiles, rubber and metal [5–7].

Among ceramic tiles, porcelain stoneware is the most diffuse product, representing 88.5% of the total Italian productions (354.5 M m$^2$ in 2019) [8]. About 15–20% of these tiles are addressed to outdoor application, especially those simulating natural stones or cement. The most common formats of ceramic tiles are both square and rectangular, while the most versatile and marketed format is the 30 cm × 60 cm.

The ceramic industry sector is a large consumer of natural resources and energy and, as a consequence, a rather large emitter of $CO_2$ [9]. More than 50% of natural resources (sands, feldspars and clays) come from foreign countries and travel by truck for 35% (62 g $CO_2$/ton km), by ship for 51% (34 g $CO_2$/ton km) and/or by train for 14% (26 g $CO_2$/ton km) of journeys [10]. The thermal energy used in the process is usually obtained from natural gas, whose combustion produces emissions of carbon dioxide. In particular, spray

drying and firing are the process phases that can be considered as mainly responsible for $CO_2$ emissions. Therefore, today, research focuses on the implementation of green hydrogen kilns by means of water electrolysis [9].

The wastes produced by the ceramic industrial process are unfired scrap tiles, fired scrap tiles, mud produced by the washing lines, lapping and polishing mud, dried grinding residues and exhausted lime by the fume abatement system. Almost all of these wastes (pre-consumer wastes) are reused in the same process, in a closed-loop cycle. Only the exhausted lime is landfill confined as dangerous waste, to avoid any risks in its reuse in a traditional tile mix (i.e., rheological risks and risks related to workers' health and safety) [11,12].

European regulations encourage and boost industries toward a green and circular economy in which "reuse" and "preparation for reuse" are the keywords to reach an innovating to zero, ideal future at zero emission, zero waste and zero non-recyclable products [13]. Finally, the transition to a circular economy is a complex process requiring wide multi-level and multi-stakeholder engagement and can be facilitated by appropriate policy interventions. Taking into account the importance of a well-balanced policy mix, which involves a variety of complementary policy instruments, the Circular Economy Action Plan of the European Union [14] includes a section on "getting the economics right" in which it encourages the application of economic instruments.

In the last few years, with the target of demonstrating a reduction in environmental impact and a significant reuse of waste materials, a new concept of traditional ceramics has been developed by replacing more natural raw materials with (i) new glassy raw materials for traditional ceramic tiles from the vitrification of different wastes able to crystallise during firing [15] and (ii) different types of waste opportunely balanced to obtain a form of waste synergy during firing [16].

The use of several kinds of waste in traditional ceramic tiles has been investigated since the late 1990s. Many works study the recycling of different wastes in traditional ceramic materials, especially porcelain stoneware tiles [17]. Waste glass (soda-lime glass) in urban collection is one of the most widely recycled forms of waste employed in traditional ceramic tile formulations, but until now it was not possible to recycle a high percentage of material (more than 20%) at an industrial level. In Italy the amount of glass from urban separate collection is more than 1.5 M ton/year. About 70% of this amount is reused by the glass industry in a closed-loop cycle. Thus, about 0.5 M ton/year are available for other uses, such as ceramic tiles production.

Unfired scrap tiles are generated after the shaping process, during the handling of unfired tiles on roller conveyor belts, in an amount of about 4% of the total production and they are generally reused in the productive cycle very easily.

This work focuses on innovation in mix design for ceramic tiles, using recycled materials with the aim of identifying the pathway towards highly recycled ceramic tiles. As shown in Figure 1, this path passes through four milestones:

1. "Applied research" demonstrates the feasibility of work, also considering the waste availability. Since 2000, many scientific papers have been focused on porcelain stoneware tiles and about 20% of them dealt with waste recycling. A search on the Scopus website [18], using "porcelain stoneware" as a keyword, showed more than 400 papers and about 140 of them are about "waste recycling". Most of these scientific papers (more than 60) were made in Italy (see Figure 2);

2. The "pilot" scale consists of a technological transfer from the laboratory to the industrial level and the industrial pilot production (a relatively small production carried out in the industrial plant). For the state of the art of the pilot milestone, the patents results based on the Free Patents Online website [19] are shown in Figure 3. In the last 20 years, 112 patents have been registered concerning the ceramic waste recycling. Most of them are European (61) and fewer have international scope (51). However, after a deeper analysis of these registered patents, only four of them deal with material recycling in porcelain bodies [20–23];

3. The "scale-up" milestone includes actual industrial production, product certification and market uptake. For the scale-up milestone state of the art, projects close to the market were considered. In particular, the Life EU project database [24] and the Eco-Innovation EU projects database [25] were checked. These databases showed 23 projects on ceramic tiles and glazes (16 led by Italian partners and 7 by Spanish ones) but only five projects deal with ceramic tiles and recycling. Three of them are not focused on porcelain stoneware tiles [26–28] but deal with ceramic wall tiles or glazed ceramic tiles for outdoor use. The other two projects [29,30] are both already finished. The GLASS PLUS project [29] was about recycling of 20% of the glass from the cathode ray tubes of dismantled televisions. The other one, WINCER [30], aimed to recycle at least 70% of pre- and post-consumer waste, while it reached 85% of recycled materials;

4. Industrial "practice" becomes a reality when all the previous steps have been successfully achieved and the pathway is consolidated. In this last phase, the concrete benefits of sustainability (social, environmental and economic) can be reached. As an example of path towards concrete innovation in tile mix design, the WINCER project was selected for its ambitious targeted results with undeniable sustainability benefits. This project, completed in December 2017, demonstrated the feasibility of industrially producing high quality ceramic tiles using pre- and post-consumer wastes, thus replacing a huge amount (85 wt%) of natural raw materials.

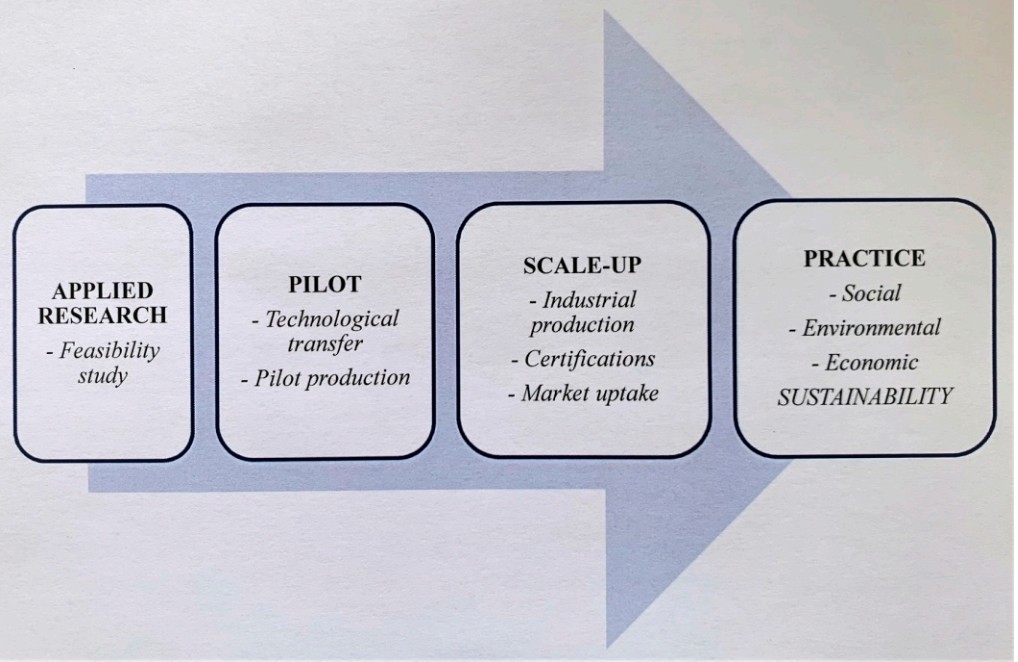

**Figure 1.** Conceptual scheme of the path towards ceramic tiles with a high content of recycled materials.

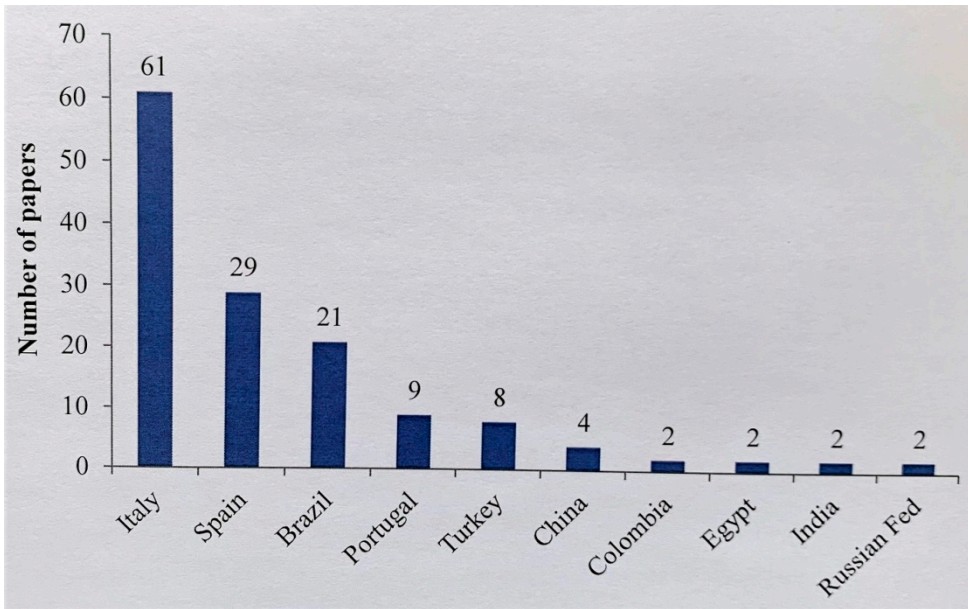

**Figure 2.** Geographical origin of scientific papers on waste recycling in porcelain stoneware mixes, since 2000.

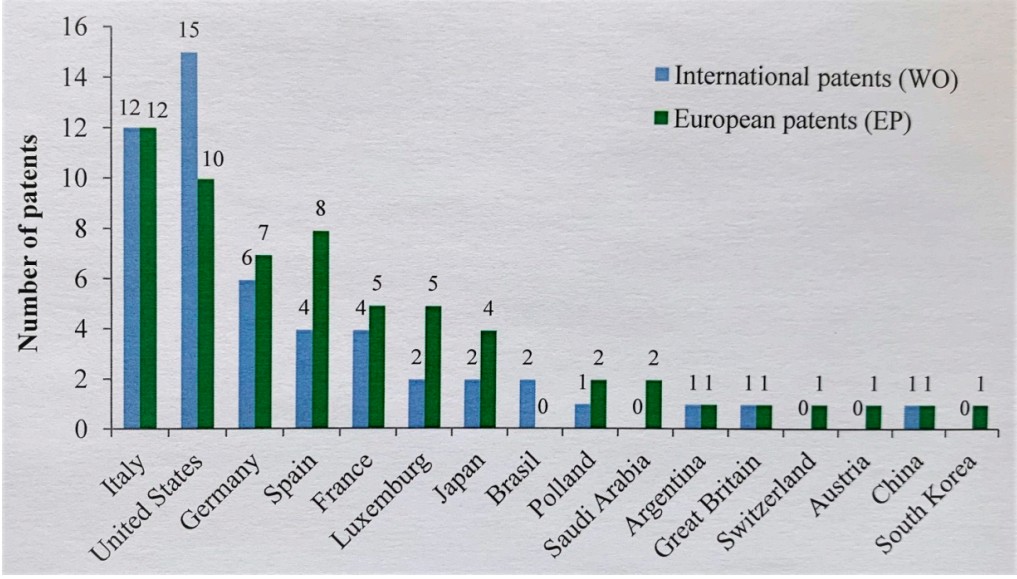

**Figure 3.** Geographical origin of patents on waste recycling in ceramic mixes, since 2000.

## 2. Materials and Methods

In the frame of applied research, a new concept of ceramic tile mix has been developed. The natural plasticizing agents (clays), fluxing agents (feldspars) and tempering agents (sands) are substituted by pre- and post-consumer wastes. This High Recycled Content (HRC) body contains 85 wt% of recycled materials (soda-lime scrap glass and unfired scrap tiles), and 15 wt% of natural clays.

An average mix of traditional porcelain stoneware consisting of 43 wt% of kaolinitic and illitic clays and 57 wt% of feldspathic sands, was chosen as reference material.

These two mixes, HRC and traditional, were prepared in the laboratory by milling the raw materials in a centrifugal ball mill (M.M.S., Nonantola, Italy) for 1 h, with 35 wt% water and 0.5 wt% deflocculating agent, FLUICER PD73 (Ceramco, Conway, NH, USA, Zschimmer & Schwarz Group, Lahnstein, Germany). To obtain powders suitable for

shaping, the slips were dried overnight in an oven at 110 °C, crushed by hand in an agate mortar and sieved through 125 μm mesh.

The chemical composition of the 85% HRC and of the traditional mixes was determined using inductively coupled plasma emission spectroscopy (ICP-OES Optima 3200 XL, PerkinElmer, Waltham, MA, USA). The quantitative mineralogical composition was also determined by X-ray diffraction analysis (PW3830, Panalytical, Almelo, Netherlands). Powdered specimens were side loaded to minimise the preferred orientation. Data were collected in the angular range 3–80° 2θ, in steps of 0.02° and 5 s/step, and Rietveld refinements were performed using GSAS-EXPGUI software.

Specimens, in form of disks and bars, were prepared by adding 6 wt% water to the dried powders, followed by uniaxial pressing at 40 MPa (P.I.L., Sacmi, Imola, Bologna, Italy). Sintering was carried out in an electric laboratory furnace (FM76, Forno Mab Srl, Rozzano, Italy), following scheduled thermal cycles, reaching six different maximum temperatures, adopting a heating rate of 5 °C/min and a natural cooling at room temperature. The sintering behaviour of the fired specimens was evaluated on the basis of their linear shrinkage and water absorption according to the recommended test method for ceramic tiles, reported in EN ISO 10545-3 [31]. The optimum firing temperature of porcelain stoneware material corresponds to a low water absorption ($\leq$0.5 wt%) and to a maximum linear shrinkage (7–8%).

The quantitative mineralogical compositions of the fired specimens were determined using the Rietveld-RIR method to also estimate the amount of amorphous phase. In this case, the fired and powdered specimens were diluted with 10 wt% of alumina powder SRM 676a (NIST, Gaithersburg, MD, USA) as internal standard and data were collected in the angular range 10–80° 2θ, with steps of 0.02° and 5 s/step.

To assess the degree of synergy of raw materials during firing, the crystalline index was calculated according to Equation (1) which considers the ratio (%) of new crystalline phases formed during sintering, compared to the amorphous phase and residual crystals as relicts coming from unmelted raw materials.

$$Crystalline\ Index = \frac{\%\ New\ Crystals}{\%\ Amorphous\ Phase + \%\ Residual\ Crystals} \times 100 \qquad (1)$$

The microstructure of the fired materials was observed by a scanning electron microscope, SEM (Zeiss EVO 40, Jena, Germany) equipped with an energy dispersion X-ray attachment, EDS (Inca, Oxford Instruments, Abingdon, UK), observing suitable specimens polished to a mirror-like finish using a lapping machine (LECO VP-150, St. Joseph, MI, USA) and etched using a 1% HF solution for 1 min.

Pyroplastic deformation was measured using a previously developed method in which the span is adjusted to keep the bending on the sample nearly constant [32]. Five bar specimens, $70 \times 10 \times 6\ mm^3$, were tested at the optimum firing temperature at a bending stress of 40 kPa for both the compositions (85% HRC and traditional porcelain stoneware). The unfired bars were placed on separated gauge blocks at an appropriate span to obtain the initial height (Hi). After firing, the bars were placed on the same gauge blocks, the height measured again (Hf), and the linear shrinkage (S) of the bars was measured. The pyroplastic deformation, typically reported in mm, is calculated from Equation (2).

$$Pyroplastic\ Deformation = Hi - Hf - S \qquad (2)$$

The closed porosity was determined by image analysis (LEICA, LAS v. 3.8, Wetzlar, Germany) on the basis of at least ten digital images acquired by an optical microscope (OM, Leica DM-LM, Wetzlar, Germany) for each fired sample.

The flexural strength of the fired bar specimens, $65 \times 9 \times 5\ mm^3$, was measured using a universal testing machine (10/M, MTS, Eden Prairie, MN, USA), equipped with a three-point bending apparatus, 60 mm roller span, with a crosshead speed of 5 mm/min. The modulus of elasticity was also evaluated by means of an extensometer applied in

the middle of the surface of the bars subjected to the tensile stress. The average flexural strength, $\sigma$, was calculated from twenty correctly fractured specimen results and Weibull's modulus, $m$, was evaluated by the least squares method and linear regression analysis, using $Pn = (I - 0.5) / N$ as the probability estimator.

On a pilot scale, the 85% HRC mix was prepared following the traditional ceramic tile process. The production process starts with the mixing of 15 wt% natural clay raw materials and 85 wt% pre- and post-consumer waste (glass scrap and unfired scrap tiles), in a continuous milling machine with water to form the ceramic suspension. This step includes the addition of fluidifying agent to improve the stability of the slurry. The slurry produced is then sent to spray drier capable of producing a granulated powder suitable for pressing to shape the tiles (30 cm $\times$ 60 cm). These tiles were then dried at about 200 °C. The industrial process for the firing phase of the 85% HRC tiles reaches the maximum temperature of 1025 °C (instead of 1200–1250 °C as for a traditional production of porcelain stoneware tiles) with duration of 39 minutes from cold to cold (duration similar to a traditional process). The fired tiles were tested in laboratory to verify the water absorption and the other technical parameters able to classify these tiles as porcelain stoneware belonging to class BIa (EN 14411) [33].

The actual industrial production of 30 cm $\times$ 60 cm tiles was then performed. UNI Keymark certification was obtained, once the requirements of EN 14411 had been satisfied, according to the standard test methods. In particular, the following performances were evaluated: determination of dimensions and surface quality (ISO 10545-2) [34], determination of water absorption, apparent porosity, apparent relative density and bulk density (ISO 10545-3) [31], determination of modulus of rupture and breaking strength (ISO 10545-4) [35], determination of impact resistance by measurement of coefficient of restitution (ISO 10545-5) [36], determination of resistance to surface abrasion for glazed tiles (ISO 10545-7) [37], determination of linear thermal expansion (ISO 10545-8) [38], determination of moisture expansion (ISO 10545-10) [39], determination of frost resistance (ISO 10545-12) [40], determination of chemical resistance (ISO 10545-13) [41], determination of stain resistance (ISO 10545-14) [42], determination of lead and cadmium release (ISO 10545-15) [43].

From the point of view of the practice milestone, the sustainability performances were evaluated on the basis of the three pillars, environmental, social and economic. To assess the environmental performance of innovative tiles, the Life Cycle Assessment (LCA) study was performed following the "cradle-to-gate" criterion and according to ISO 14040 [44] and ISO 14044 [45]. The modules of the production life cycle are those related to the production stages:

A1. Raw materials supply (extraction, processing, recycled material);

A2. Transport to manufacturer;

A3. Manufacturing.

The environmental impact assessment categories/indicators considered are those required by the Environmental Product Declaration (EPD) following the EN 15804 [46]:

- Photochemical Ozone Creation Potential (POCP): ozone formation in the lower atmosphere causing summer smog;
- Ozone Depletion Potential (OPD): ozone depletion in the higher atmosphere;
- Global Warming Potential (GWP): greenhouse gases causing climate change;
- Eutrophication Potential (EP): emissions causing over-fertilisation of soil or water;
- Acidification Potential (AP): emissions causing acidifying effects (acid rain, forest decline);
- Abiotic Depletion Potential fossil (ADPf): scarcity of resources (fossil energy carriers);
- Abiotic Depletion Potential elementary (ADPe): scarcity of resources (ores, silicates).

The reduction in Respirable fraction of Crystalline Silica (RCS) [47,48] was estimated in comparison with traditional production, using the SWERFCS (Size Weight Respirable Fraction of Crystalline Silica) method. It is based on the experimental determination of RCS by sedimentation of powder in double-distilled water, following the EN 481 [49].

Economic evaluations on the costs of the 85% HRC ceramic tiles, with respect to a traditional production, were carried out by comparing the costs of raw materials and process costs to prepare both mixes for porcelain stoneware tiles.

## 3. Results and Discussion

### 3.1. Applied Research

The chemical and mineralogical compositions of the 85% HRC and traditional porcelain stoneware bodies are shown in Tables 1 and 2. While the chemical compositions are not significantly different, apart a lower amount of alumina and a higher amount of lime and soda in the 85% HRC body, the main difference concerns the mineralogical composition, where a higher amount of amorphous phase is present in the 85% HRC body. On the other hand, quartz and plagioclase are significantly lower than in the traditional porcelain stoneware body. The presence of soda-lime glass cullet, which replaces the natural feldspathic sands in the 85% HRC body, justifies this composition.

**Table 1.** Chemical compositions (oxides wt%) of the unfired powders of the 85% HRC mix and traditional porcelain stoneware mix.

|  | 85% HRC Mix | Traditional Porcelain Stoneware Mix |
| --- | --- | --- |
| Loss on ignition | 2.92 | 3.88 |
| $SiO_2$ | 74.30 | 70.70 |
| $Al_2O_3$ | 10.51 | 18.58 |
| $TiO_2$ | 0.13 | 0.62 |
| $Fe_2O_3$ | 0.67 | 0.75 |
| CaO | 4.43 | 0.69 |
| MgO | 0.47 | 0.39 |
| $K_2O$ | 1.23 | 2.80 |
| $Na_2O$ | 5.34 | 1.59 |

**Table 2.** Mineralogical compositions (wt%) of the unfired powders of the 85% HRC mix and traditional porcelain stoneware mix.

|  | 85% HRC Mix | Traditional Porcelain Stoneware Mix |
| --- | --- | --- |
| Quartz | 12.0 | 27.3 |
| Illite | 8.1 | 15.8 |
| Kaolinite | 7.7 | 12.3 |
| Plagioclase | 15.2 | 37.6 |
| Microcline | 2.4 | 4.5 |
| Amorphous phase | 54.6 | 2.5 |

Figure 4 shows graphically the water absorption and linear shrinkage values at different temperature for both 85% HRC and traditional materials. The optimum firing temperatures, represented by the lowest water absorption (less than 0.5 wt%) and the highest linear shrinkage (approximately 7–8%), are 960 and 1160 °C for the 85% HRC and the traditional porcelain stoneware, respectively.

The quantitative mineralogical compositions of the samples fired at their optimum temperature are shown in Table 3 together with the crystalline index for both the compositions. Although the 85% HRC material is fired at a significantly lower temperature than a traditional porcelain stoneware tile composition (200 °C lower), it is interesting to note that the crystalline index is significantly higher (30% instead of 6% for a traditional mix).

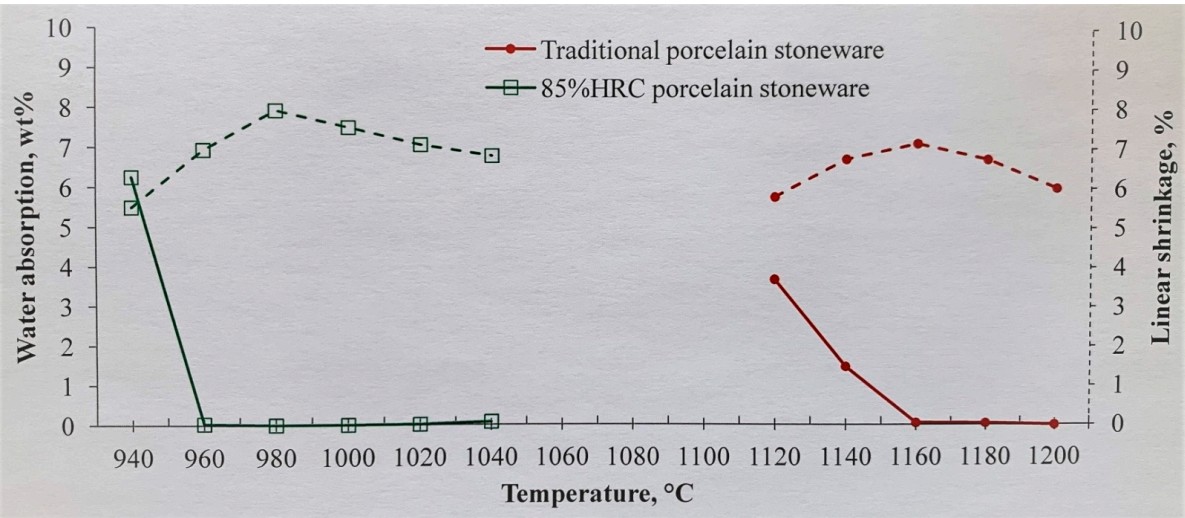

**Figure 4.** Water absorption (solid lines) and linear shrinkage (dashed lines) of 85% HRC and traditional porcelain stoneware samples fired in a laboratory electrical furnace.

**Table 3.** Quantitative mineralogical compositions (wt%) and crystalline index (%) of 85% HRC and traditional porcelain stoneware materials sintered in the laboratory furnace.

|  | 85% HRC Porcelain Stoneware Fired at 960 °C | Traditional Porcelain Stoneware Fired at 1160 °C |
|---|---|---|
| Quartz | 7.6 wt% [1] | 21.3 wt% [2] |
| Plagioclase | 14.8 wt% [2] | 9.4 wt% [2] |
| Mullite | - | 5.4 wt% [1] |
| Wollastonite | 8.5 wt% [1] | - |
| Amorphous phase | 69.1 wt% | 63.9 wt% |
| Crystalline index | 30 % | 6 % |

[1] Residual crystals from unmelted raw materials. [2] New crystals formed during sintering.

It is known from the literature that the presence of soda-lime glass in porcelain stoneware bodies allows the crystallisation of plagioclase and wollastonite as new phases [50]. The synergy of the waste during firing leads to a higher incipient crystallisation, than in traditional porcelain stoneware, where only mullite crystallises from about 1050 °C [51]. Due to the fact that the 85 % HRC material is fired at lower temperature (960 °C), mullite was not detected (Table 3). Figure 5 shows the micrographs of the polished and etched surfaces of both materials fired at their optimum sintering temperatures. In the 85% HRC sample fired at 960 °C, crystalline phases such as wollastonite, with a rather elongated and thick shape, and plagioclase, with a pseudospherical shape, can be observed (Figure 5a). In the traditional porcelain stoneware sample fired at 1160 °C, elongated and thin mullite needles are clearly visible (Figure 5b).

Table 4 shows the main characteristics of 85% HRC and traditional porcelain stoneware materials sintered in the laboratory furnace. The pyroplastic deformation during firing is an important parameter to understand the firing behaviour of the material. A low pyroplastic deformation (indicatively <0.8 mm) indicates a negligible risk of tile deformation during industrial firing. The values reported in Table 4 are very similar for both the compositions, traditional and 85% HRC. The mechanical properties, flexural strength and Weibull's modulus are all quite similar for 85% HRC fired at 960 °C and traditional porcelain stoneware fired at 1160 °C; Young's modulus is lower in the 85% HRC material and it can be associated with the slightly higher amount of closed porosity which results in a less rigid ceramic matrix than in the traditional composition. In fact, the combination of different wastes in the 85% HRC batch, allowing incipient crystallisation during firing, gives rise to a material with similar or improved mechanical properties than the traditional

one, even if the maximum sintering temperature is reduced. This mechanism is able to overcome the risk associated with dimensional instability during firing; it is therefore possible to produce even relatively large ceramic tiles on an industrial scale. These results indicate that the innovative composition for porcelain stoneware is suitable for technology transfer on an industrial scale.

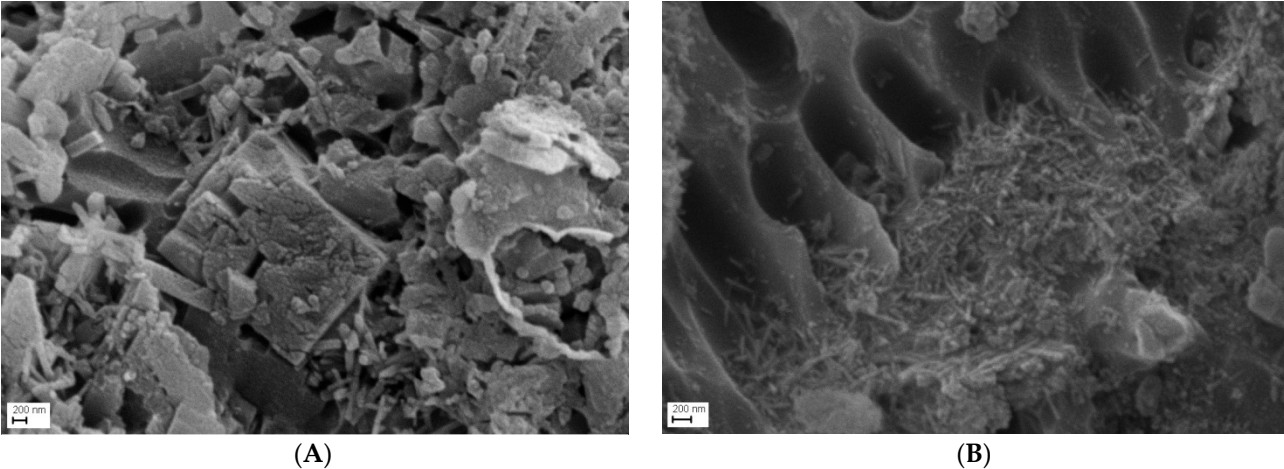

(**A**) (**B**)

**Figure 5.** SEM micrographs of polished and etched sample surfaces. (**A**) 85% HRC sample fired at 960 °C showing wollastonite an plagioclase crystals; (**B**) porcelain stoneware sample fired at 1160 °C showing elongated mullite crystals.

**Table 4.** Main characteristics of 85% HRC and traditional porcelain stoneware materials sintered in the laboratory furnace.

|  | 85% HRC Porcelain Stoneware Fired at 960 °C | Traditional Porcelain Stoneware Fired at 1160 °C |
| --- | --- | --- |
| Pyroplastic deformation | 0.5 ± 0.2 mm | 0.4 ± 0.3 mm |
| Closed porosity | 14 ± 3% | 9 ± 3% |
| Flexural strength | 80 ± 2 MPa | 83 ± 2 MPa |
| Young's modulus | 69 ± 1 GPa | 57 ± 1 GPa |
| Weibull's modulus | 15 | 16 |

*3.2. Pilot Scale*

A pilot production of about 100 m$^2$ of 85% HRC 30 cm × 60 cm tiles was carried out on an industrial scale.

The production process is shown schematically in Figure 6, in comparison with the production of traditional porcelain stoneware tiles. While the steps of the production process are the same between an industrial production of traditional porcelain stoneware and one of 85% HRC porcelain stoneware, with the new ceramic body it is possible to reach a significant saving of natural resources due to the total substitution of feldspathic sands (100% saving) and the partial substitution of clays (more than 60% saving). In the milling phase, energy savings of about 20% are expected for the production of 85% HRC compared to the traditional ceramic tile process, because the milling of scrap glass is easier than the milling of quartz grains contained in the siliceous feldspathic sands. Finally, the maximum sintering temperature of 85% HRC ceramic tiles is about 200 °C lower in comparison to traditional porcelain stoneware tiles; therefore, a 10% saving of methane and, consequently, about 10% less $CO_2$ emissions has been estimated. Interestingly, the industrial production of porcelain stoneware tiles requires higher firing temperatures than a laboratory firing, respectively 1210 versus 1160 °C for traditional porcelain stoneware tiles and 1025 versus 960 °C for 85% HRC tiles. This is due to the different temperature gradient of the industrial fast firing process (39 minutes from cold to cold) compared to the laboratory slow firing schedule (approximately 18–20 h from cold to cold).

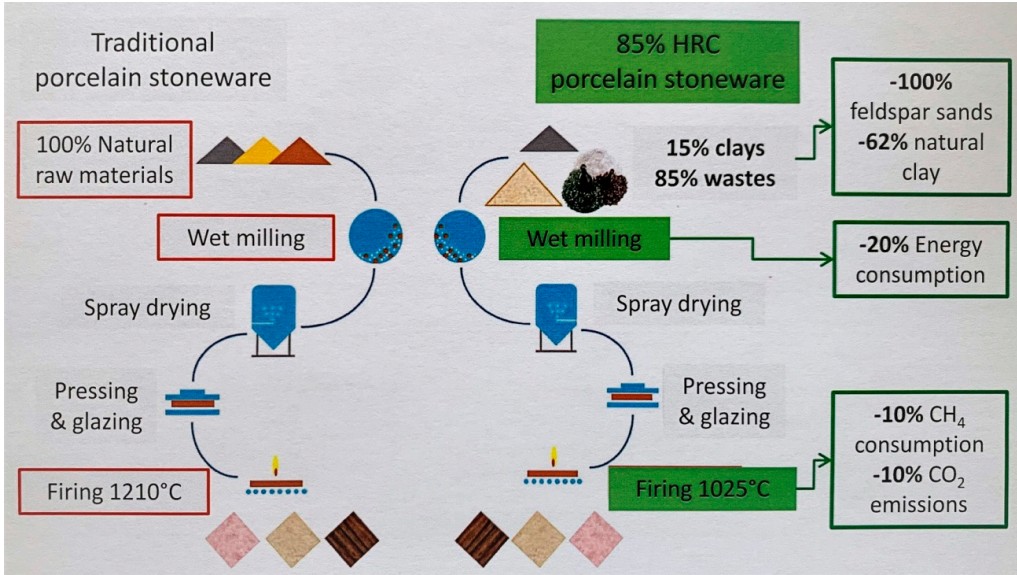

**Figure 6.** Comparison of the industrial production process steps and envisaged savings.

The 85% HRC fired tiles belong to the same class of porcelain stoneware product (class BIa of the EN 14411 classification) with a water absorption of less than 0.5 wt%.

### 3.3. Industrial Scale-Up and Practice

The industrial production of about 10,000 m² of 85% HRC 30 cm × 60 cm ceramic tiles was performed. The tiles coming out of the industrial kiln are shown in Figure 7.

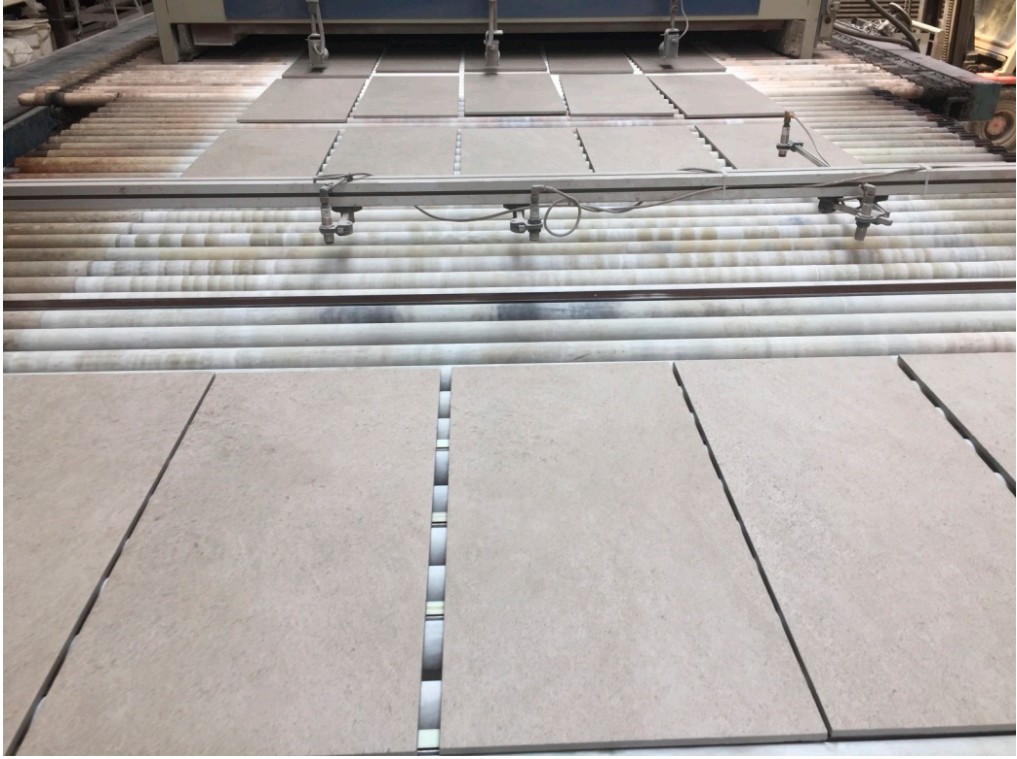

**Figure 7.** Industrial production of 30 cm × 60 cm tiles.

This product belongs to class BIa and all technical performances were evaluated. The results are shown in Table 5, together with the requirements specified in the Annex G of

EN 14411. This product fulfilled all these requirements and obtained the UNI Keymark quality certification.

**Table 5.** Technical performances of 85% HRC porcelain stoneware tiles produced on an industrial scale.

| Standard | Description | Sampling | Requirement | Result |
|---|---|---|---|---|
| ISO 10545-3 | Water absorption | 5 specimens 20 cm × 20 cm cut from 5 tiles | E(b) ≤ 0.5% Max single value 0.6% | E(b) = 0.1% Max single value 0.1% |
| ISO 10545-2 | Dimension | 7 whole tiles 30 cm × 60 cm | Length and wideness ± 0.6% Thickness ± 5% Straightness of edges ± 0.5% Orthogonality ± 0.5% Surface quality 95% without defects | −0.3% −2.9% 0.1% −0.2% 100% without defects |
| ISO 10545-4 | Flexural strength | 7 whole tiles 30 cm × 60 cm | Breaking strength > 1300 N Flexural tensile strength > 35 N/mm$^2$ | 1599 N 42.3 N/mm$^2$ |
| ISO 10545-5 | Impact resistance | 5 specimens 7.5 cm × 7.5 cm cut from 5 tiles | Coefficient of restitution > 0.55 | 0.73 |
| ISO 10545-7 | Abrasion resistance | 11 specimens 10 cm × 10 cm | Declared value (Min. class 4) | Class 4 |
| ISO 10545-8 | Coeff. of expansion | 2 specimens 2.5 × 0.5 × 0.5 cm$^3$ | Declared value | 7.9–8.2×10$^{-6}$ °C$^{-1}$ |
| ISO 10545-10 | Humidity expansion | 3 whole tiles 30 cm × 60 cm | Declared value (max. 0.6 mm/m) | 0.1 |
| ISO 10545-12 | Frost resistance | 10 whole tiles 30 cm × 60 cm | Resistant (0 tiles with defects) | 0 tiles with defects |
| ISO 10545-13 | Chemical resistance | 3 specimens 5 cm × 5 cm cut from 3 tiles | Low conc. acid and base: declared High conc. acid and base: declared Household chemicals and swimming pool salts: min. class B | Class A Class A Class A |
| ISO 10545-14 | Stain resistance | 5 specimens 15 cm × 15 cm cut from 5 tiles | Chrome green oil: min class 3 Iodine solution: min class 3 Olive oil: min class 3 | Class 5 Class 5 Class 5 |
| ISO 10545-15 | Pb and Cd release | 3 specimens 15 cm × 15 cm cut from 3 tiles | Declared value (food contact Dir. 2005/31/CE: max 0.8 mg/dm$^2$) | <0.001 mg/dm$^2$ |

Before the industrial practice milestone of the HRC path, it is important to consider that, for technical reasons, this type of production must be carried out in a dedicated industrial line, which is not shared with a traditional porcelain stoneware production due to the significantly different sintering temperature.

Today, the industrial approach is to implement production concepts for small series, on demand and just in time, where production is according to the needs and tastes of the market, varying in a great number of sizes, thicknesses and surfaces (from smooth to structured, with different tones). The main reason for such discontinuous production lies in the limited choice of glazes for the HRC body. Glazes for traditional porcelain stoneware tiles are suitable for firing at relatively high temperatures (1200–1250 °C) and, after firing, are characterised by excellent technical performances that do not alter the performances of the porcelain stoneware material. A wide variety of glazes for HRC porcelain stoneware tiles, with high surface properties such as stain, chemical and abrasion resistance, have not yet been implemented. For this reason, the choice is limited to a few products able to

guarantee high surface performance despite the fact that sintering takes place at relatively low temperatures (around 1000 °C).

It is expected that sustainability benefits (environmental, social and economic) can be achieved during the industrial practice milestone. The results of the LCA study demonstrated the environmental advantages of producing 85% HRC porcelain stoneware production compared to a production of traditional porcelain stoneware. In particular, Figure 8 shows the relative contribution of each life cycle stage to different environmental indicators. The main contribution to the total impact of production, from cradle-to-gate processes, is due to the raw materials supply module, A1, for each impact indicator. This module takes into account both energy and the production of raw materials (such as clay, chemical compounds, pigments, etc.). The transport module, A2, has significant contributions to two indicators, EP and AP, and it is due to both the production of fuel and the emissions associated with its combustion. The global production module, A3, contributes to EP, POCP and ODP due to the production of paper/cardboard for packaging. APDe is mainly influenced by inks and ADPf by energy production processes, both belonging to the A1 module.

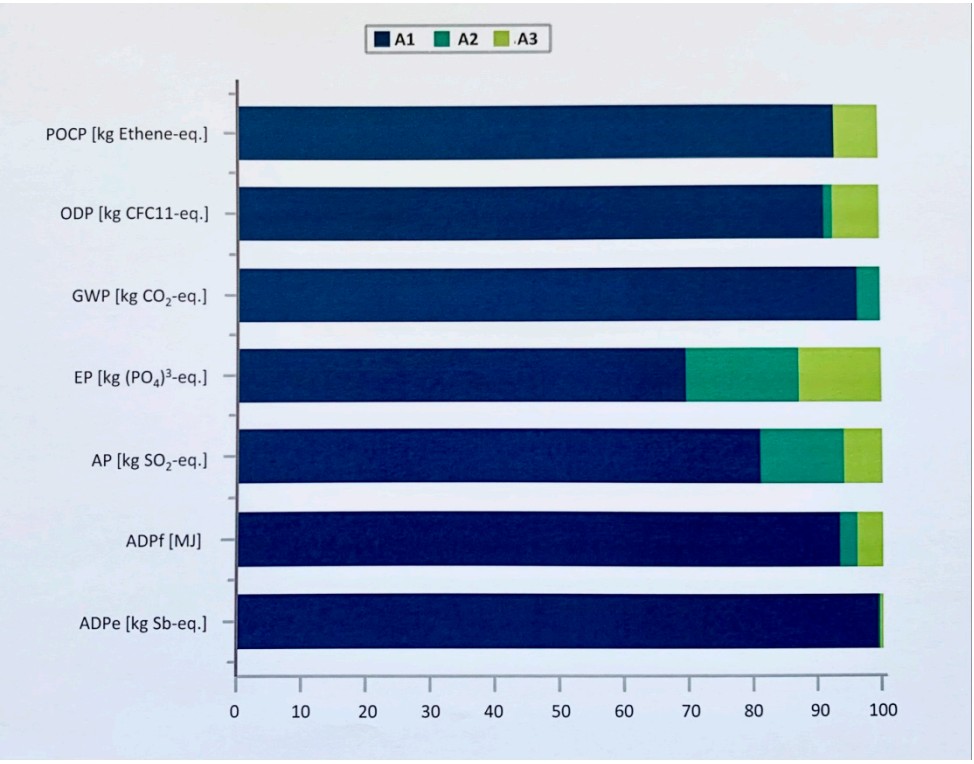

**Figure 8.** Relative contribution (%) of each life cycle stage, A1, A2 and A3 (raw materials supply, transport and manufacturing, respectively) to different environmental indicators for 85% HRC product.

A comparison of the impacts of the environmental indicators was made between 85% HRC porcelain stoneware and a traditional porcelain stoneware. Table 6 shows the relative contributions, of both types of tiles, as a percentage of the contribution of the industrial production of spray-dried powder (only the raw materials preparation was considered) on the overall manufacture A1, A2 and A3 (A1 + A2 + A3 is 100%). In the case of the 85% HRC tile, the relative contribution of the spray-dried powder production is much lower than that of the traditional porcelain stoneware tile.

**Table 6.** Relative contribution of environmental impact indicators between 85% HRC spray-dried powder production and an average traditional spray-dried powder production, as a percentage of the overall contribution of the A1, A2, A3 manufacturing process (raw materials supply, transport and manufacturing, respectively).

| Environmental Indicators (EN 15804) | 85% HRC Porcelain Stoneware | Traditional Porcelain Stoneware |
|---|---|---|
| GWP [kg $CO_2$-eq.] | 1.09 % | 24–25 % |
| ODP [kg CFC11-eq.] | 0.43 % | 69–75 % |
| AP [kg $SO_2$-eq.] | 2.80 % | 54–56 % |
| EP [kg $(PO_4)^3$-eq.] | 3.35 % | 26–27 % |
| POCP [kg Ethen-eq.] | 2.08 % | 37–39 % |

The social benefits relate to health and safety in the workplace. The RCS potential of the spray-dried powders of the 85% HRC mix is reported in Table 7 in comparison with a traditional porcelain stoneware mix. Due to the significantly lower amount of quartz in the mix (about 57 wt% less), the HRC spray-dried powder is characterised by a lower RCS value of about 63% with respect to a traditional spray-dried powder. In any case, in industrial practice, both of these values give rise to significantly lower exposure limits (0.1 mg/m$^3$ measured over a reference period of 8 h) [47,52].

**Table 7.** Respirable fraction of Crystalline Silica (RCS) of the traditional and 85% HRC spray-dried powders.

| | 85% HRC Mix | Traditional Porcelain Stoneware Mix |
|---|---|---|
| RCS potential | 1.9% | 5.2% |

The economic benefits can be related to the lower industrial costs required to produce the 85% HRC body (raw material and milling costs). In Table 8 the costs to produce the 85% HRC body are compared with those needed to produce an average body of traditional porcelain stoneware batch. For pre-consumer waste (unfired tile scrap), a cost of EUR 1/ton was considered because this scrap consists of raw materials that have already been processed (milled, dried and pressed) and are reintroduced into the ceramic process in a closed-loop cycle. For post-consumer waste (soda-lime glass scrap), a relatively high cost of EUR 30/ton was considered, due to the fact that this is processed glass which has become a secondary raw material and can be useful as flux for industrial applications (not only ceramics).

**Table 8.** Average costs to produce 1 m$^2$ of HRC mix and 1 m$^2$ of average mix of a traditional porcelain stoneware tile (the firing phase was not considered).

| | 85% HRC Tile | | | Traditional Porcelain Stoneware Tile | | |
|---|---|---|---|---|---|---|
| | Composition | Cost/ton | Cost/m² | Composition | Cost/ton | Cost/m² |
| Clays | 3.3 kg | 79 € | 0.26 € | 7.92 kg | 69 € | 0.55 € |
| Feldspars and sands | - | | | 14.08 kg | 38.9 € | 0.55 € |
| Pre-consumer waste | 6.6 kg | 1 € | 0.0066 € | - | | |
| Post-consumer waste | 12.1 kg | 30 € | 0.363 € | - | | |
| Chemicals | | | 0.002 € | | | 0.002 € |
| Milling | | 25 € | 0.55 € | | 30 € | 0.66 € |
| TOTAL | 22 kg | | 1.18 € | 22 kg | | 1.76 € |

The economic saving is about 33%. Moreover, considering the firing step, the saving increases further due to the lower methane consumption (about −10%).

## 4. Conclusions

The pathway from pilot to practice, through scale-up and sustainability goals, has been a challenge for many global sustainability interventions. Scale-up and institutionalisation of successful practices are the aims of every pilot project. Proven benefits on a small scale can generate a measurable impact on sustainability. In particular, the main benefits are:

- Acquiring world leadership in waste-based ceramic materials. The use of at least 85% recycled materials strengthens the waste market, which becomes a valuable resource and helps preserve natural stocks of virgin and important minerals such as clays, limestone and feldspar and also reduces imports of minerals such as zirconia, bauxite and magnesia from overseas;
- Widening the market for more sustainable ceramic products to replace other materials such as concrete, granite and marble. These innovative tiles are not only able to minimise the environmental impact of the extracting natural materials in the quarries or the felling trees, but also divert waste from landfills and reuse industrial scraps;
- Reducing in the energy consumption of the milling and firing processes (electricity and methane are the highest factors of impact on the production cost of tile);
- Improving health in the workplace thanks to the lower amount of free crystalline silica in the ceramic body.

The innovative tiles developed at national level will give a boost for further fruitful international activities to develop sustainable, cost effective, structural ceramics from the "waste synergy". As environmental legislation and EU policies will become more and more restrictive in the coming years, the obtained results are very useful for companies that will be required to comply with regulations for recycling wastes, which are currently dumped.

Further research should focus on glazes suitable for this type of porcelain stoneware, which is fired at low temperature. This will make it possible to offer a greater choice of sustainable products of high technical and aesthetic quality.

**Funding:** This research was funded by the Executive Agency for SMEs—EASME—in the frame of the European project "Waste synergy in the production of innovative ceramic tiles"—WINCER—(http://www.wincer-project.eu/ (accessed on 19 July 2021)), grant number ECO/13/630426.

**Institutional Review Board Statement:** Not applicable.

**Informed Consent Statement:** Not applicable.

**Data Availability Statement:** Most of the data can be found here: http://www.wincer-project.eu/ (accessed on 19 July 2021) and here: http://www.wincer-project.eu/wp-content/uploads/2018/01/Layman-Report-Wincer-ENG_DEF-min.pdf (accessed on 19 July 2021). Some data presented in this study are only available upon request to the corresponding author.

**Acknowledgments:** The author thanks all colleagues at Centro Ceramico for their professional technical support. This publication would not have been possible without the precious collaboration of Marazzi Group S.r.l. (Sassuolo, Italy) staff, in particular Claudio Beneventi, Lorenzo Valeriani and Linda Grandi.

**Conflicts of Interest:** The author declares no conflict of interest.

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
