# Peer review of "Pathway towards a High Recycling Content in Traditional Ceramics"

_ceramics, doi:10.3390/ceramics4030036_

Round 1
Reviewer 1 Report
Dear author,
Comments on improving your manuscript were added as comments in manuscript PDF. If possible, have the text proofread by a native speaker, as there are numerous occurrences of non-English language structure.
Best regards

Author Response
I thank the Reviewer for the suggestions.
I replied to all comments on improving the manuscript directy in the attached PDF file.
I also improved the English due to numerous occurrences of non-English language structure, as the reviewer highlighted.

Reviewer 2 Report
The article presents interesting results of a research project on the use of waste ceramics in the production of ceramic tiles. The use of ceramic waste in the production of new building materials is increasingly being researched by scientists. This is enforced by the European Union's environmental protection regulations. In the article, thanks to the use of 85% waste ceramics, a new material was obtained with similar characteristics compared to the base material. Such a production cycle results in a less negative impact on the natural environment (during production and transport), allows to reduce the level of extraction of natural ingredients and reduces the amount of waste deposited in landfills. It also influences the improvement of the working conditions of employees.
Refill:
Description of figure 4: dashed line - linear srinkage, solid line - water absorption,
To explain:
Differences in the value of Young's modulus of two materials (21% difference)
To correct:
Table 8, heading: Compositio… ..n, bold.
Author Response
I thank the reviewer for the suggestions.
Here following my reply to his comments:
COMMENT 1: Description of figure 4: dashed line - linear srinkage, solid line - water absorption,
REPLY 1: I agree with the reviewer and I added this description for Figure 4.
COMMENT 2: Differences in the value of Young's modulus of two materials (21% difference)
REPLY 2: The reviewer is right. I added in table 4 also the "closed porosity" values that is an important characteristics able to explain the differeces in the Young's modulus of the two materials.
COMMENT 3: To correct Table 8, heading: Compositio… ..n, bold.
REPLY 3: I corrected the style of Table 8 as suggested.
Reviewer 3 Report
Dear author
The work presented shows a very complete study of the possibilities in the use of high content of recycled materials in the manufacture of porcelain stoneware.
This topic has already been previously studied by various authors. However, this work, unlike the previous ones, addresses the four phases: analysis of previous studies “Applied research”, laboratory scale production "Pilot", industrial production, and analysis "Scale-up", and analysis of sustainable profits “Practice”.
The mineral composition of the HCR has been determined by X-ray diffraction. Nevertheless, the percentage of each of these residues (soda-lime scrap glass and crap tiles), as well as the origin of the crap tiles (porcelain stoneware, stoneware, porous tile, ...) would be interesting to know.
Likewise, thinking about the industrial applications of these materials in the manufacture of porcelain stoneware as a substitute for natural minerals, to have information on the available amounts of tile waste and soda lime scrap glass would be interesting to know.
This information could complete the discussion of the results concerning the applicability of the new materials in the manufacture of porcelain stoneware.
The conclusions indicate that these innovative tiles are more sustainable than other materials such as wood, however, this comparison is not justified by the study carried out or the bibliographic references uses.
Author Response
I thank the Reviewer fot the suggestions.
Here following my reply to comments:
COMMENT1: The mineral composition of the HCR has been determined by X-ray diffraction. Nevertheless, the percentage of each of these residues (soda-lime scrap glass and crap tiles), as well as the origin of the crap tiles (porcelain stoneware, stoneware, porous tile, ...) would be interesting to know.
REPLY 1: The reviewer is right. I expected this comment. Unfortunately this data is confidential. I don't have permission from the Company to share it. But I know that the mix composition is an important information for this paper. That is why I added the mineralogical composition in Table 2, from which it is possible to see the amount of amorphous phase.
COMMENT 2: Likewise, thinking about the industrial applications of these materials in the manufacture of porcelain stoneware as a substitute for natural minerals, to have information on the available amounts of tile waste and soda lime scrap glass would be interesting to know. This information could complete the discussion of the results concerning the applicability of the new materials in the manufacture of porcelain stoneware.
REPLY 2: I agree. It is an important data and I added this information in the "introduction".
COMMENT 3: The conclusions indicate that these innovative tiles are more sustainable than other materials such as wood, however, this comparison is not justified by the study carried out or the bibliographic references uses.
REPLY 3: Actually the comparison was only between traditional porcelain stoneware production and high recycled content one. The reviewer is right, in the conclusion a too strong comments about other materials is not justified. I rewrote the sentence.